# Bacterial Assessment of Stethoscopes Used by Healthcare Workers at a Tertiary Care Government Hospital in Bharatpur, Nepal

**DOI:** 10.3390/diseases11020055

**Published:** 2023-04-01

**Authors:** Sanjib Adhikari, Neetu Adhikaree, Krishna Prasad Paudel, Roshan Nepal, Birendra Poudel, Sujan Giri, Suman Khadka, Saphala Khawas, Sanjeep Sapkota, Ramesh Sharma Regmi, Komal Raj Rijal

**Affiliations:** 1Central Department of Microbiology, Tribhuvan University, Kirtipur 44618, Nepal; 2Department of Microbiology, Birendra Multiple Campus, Bharatpur 44207, Nepal; 3Department of Microbiology, Bharatpur Hospital, Bharatpur 44200, Nepal; 4Department of Microbiology, College of Medical Sciences, Bharatpur 44207, Nepal; 5Department of Orthopaedics, Bharatpur Hospital, Bharatpur 44200, Nepal

**Keywords:** stethoscope, nosocomial infection, healthcare workers, ESBL, MBL

## Abstract

The transmission of healthcare-associated infections (HCAIs) in healthcare settings is a serious challenge in the medical fraternity. Medical devices, such as stethoscopes used by healthcare workers (HCWs), are likely to harbor a considerable number of pathogenic microbes, which may result in the transmission of HCAIs. This study sought to investigate bacterial contamination of stethoscopes used by HCWs at Bharatpur Hospital, Nepal. During the study period of 3 months from December 2019 to February 2020, a total of 87 stethoscopes were examined; bacterial pathogens were isolated and identified by culture and biochemical tests, and their susceptibilities against different antibiotics were determined using standard protocols of the Clinical and Laboratory Standards Institute (CLSI). The disc diffusion method was used primarily to screen for extended-spectrum beta-lactamase (ESBL)- and metallo-beta-lactamase (MBL)-producing isolates, followed by their confirmation using cephalosporin/clavulanate combination discs and the disc potentiation methods, respectively. In addition, molecular detection of *bla*_CTX-M_ and *bla*_VIM_ genes was performed using conventional polymerase chain reaction (PCR). Of the 87 stethoscopes examined, more than a quarter (28.7%) were colonized with different pathogenic bacteria. Bacterial contamination of stethoscopes was found to be significantly associated with various factors, such as disinfecting routine, method of disinfection, and department of the hospital (*p* < 0.05). A higher rate of bacterial contamination was observed on the diaphragm of the stethoscope (12.64%) and among HCWs who overlooked hand hygiene practices (45.45%). The prevalence of methicillin-resistant *S. aureus* (MRSA) was 44.44%, and approximately half of the Gram-negative isolates (47%) were multidrug resistant (MDR). Imipenem (81.25%) and chloramphenicol (83.33%) were found to be the most effective antibiotics for Gram-negative and Gram-positive bacteria, respectively. Phenotypic screening showed that 43.75% of isolates were ESBL producers, and 18.75% were MBL producers, but *bla*_CTX-M_ and *bla*_VIM_ genes were detected in only 31.25% and 6.25% of isolates, respectively. The results of the study call for effective stethoscope disinfection practices along with the judicious use of antibiotics by HCWs in order to minimize cross-contamination, emergence of resistance, and spread of nosocomial infections in clinical settings.

## 1. Introduction

Nosocomial infections are serious health problems worldwide that are mainly acquired during hospitalizations or long visits to hospitals [1]. Some contributing factors to the persistence of nosocomial infections are the emergence of multidrug-resistant (MDR) bacteria, the immunocompromised state of some patients, and various gene transfer mechanisms of bacteria [2]. The occurrence of nosocomial infections significantly alters or affects practices in healthcare bodies. It has been reported that at least one-third of all nosocomial infections are preventable and frequently caused by organisms in the hospital environment [3]. The surfaces of hospital floors, tools, and equipment are contaminated by various pathogenic microorganisms [1].

The stethoscope is a universal and necessary tool for medical professionals that can act as a vector for the transmission of nosocomial infections. It is regularly in contact with a large number of people and thus becomes contaminated with various pathogenic microorganisms. Disinfection of stethoscopes after each use is not an established practice anywhere [4]. If the same stethoscope is used for examining another patient without disinfecting, it may impose an infection risk on all following patients [5]. Moreover, the draping of stethoscopes around the neck is a very common practice that results in cross-contamination of the diaphragm. In addition, the commensal flora and pathogenic microbes residing in the ears of healthcare professionals contaminate the earpieces of the stethoscope. There is also a high risk of transmission of multidrug-resistant microorganisms in hospital settings through the use of contaminated stethoscopes [6].

Bacterial pathogens chiefly found on stethoscope swabs include *Staphylococcus aureus*, *Enterobacteriales*, *Clostridium difficile*, *Pseudomonas* spp., *Acinetobacter* spp., *Bacillus* spp., *Corynebacterium* spp., etc. [7,8,9]. Besides these organisms, contaminated stethoscopes can act as vectors for the transmission of an array of antibiotic-resistant bacteria, such as methicillin-resistant staphylococci and gentamicin-resistant *Pseudomonas aeruginosa* [2]. The emergence of ESBL-producing microorganisms is increasing at an alarming rate, and this does not stop in current clinical settings, as various studies of clinical samples [7,8,9,10], as well as hospital equipment [11,12], have revealed a high prevalence of multidrug-resistant ESBL- and MBL-producing organisms. Beta-lactamase production is a method bacteria employ to resist drugs such as the penicillins and cephalosporins [13]. Various studies have reported that ESBL-producing bacteria can contaminate medical examination equipment (MEE), such as stethoscopes [14,15]. Exposure of the already susceptible hospitalized patient to the resident flora of the hospital environment may exacerbate the clinical condition of the patient [5]. This study was undertaken primarily to investigate the bacterial pathogens contaminating the stethoscopes used by healthcare professionals in a major tertiary care hospital in Bharatpur, Nepal, with a special focus on ESBL and MBL production in bacterial isolates.

## 2. Methodology

This cross-sectional study was conducted in Bharatpur Hospital, Chitwan, Nepal, from December 2019 to February 2020. Bharatpur Hospital is one of the major government-owned teaching hospitals situated in Bagmati Province of the country. In this study, stethoscopes used by 87 healthcare workers from different departments of the hospital were included for bacterial investigation. After obtaining informed consent from each participant, an anonymous study questionnaire was administered to gather information on demographics, hand-washing and sanitizing habits, stethoscope usage, and handling and maintenance practices. Samples were collected from four different parts of the stethoscope, namely the right earpiece, left earpiece, bell, and diaphragm, using a sterile cotton swab aseptically. The swab was inserted into transport medium immediately after sampling and transported to the laboratory for microbial analyses.

### 2.1. Culture and Identification of Isolates

The collected swabs were inoculated onto plates of blood agar medium and MacConkey agar medium separately for each sample, incubated at 37 °C aerobically for 24 h, and examined for bacterial growth according to standard protocols [16]. Identification of bacterial isolates was performed based on their morphological and biochemical characteristics [16].

### 2.2. Antibiotic Susceptibility Testing

Antibiotic susceptibility testing (AST) was performed following the modified Kirby Bauer disc diffusion method using CLSI guidelines (2016) as a reference [17]. A total of 13 different commonly prescribed antibiotics (tetracycline (30 µg), imipenem (10 µg), chloramphenicol (30 µg), ciprofloxacin (5 µg), gentamycin (10 µg), azithromycin (15 µg), methicillin (5 µg), ceftazidime (30 µg), cefotaxime (30 µg), cefepime (30 µg), amikacin (30 µg), aztreonam (30 µg), and levofloxacin (5 µg)) procured from Hi-Media, India, were used for susceptibility testing.

### 2.3. Screening of ESBL and MBL Producers

Primary screening of ESBL producers was performed using the disc diffusion method with ceftazidime (CAZ) (30 µg) and cefotaxime (CTX) (30 µg) discs (Hi-Media, Thane, India). If the zone of inhibition was 22 mm for CAZ and/or 27 mm for CTX, the isolate was considered a potential ESBL producer, as recommended by NCCLS [18]. The combination disc method [19] was used to confirm ESBL-producing isolates in which CTX and CAZ (30 µg), alone or in combination with clavulanic acid (CA) (10 µg), were used. An increase in the ZOI of 5 mm for either antimicrobial agent tested in combination with CA versus its zone when tested alone confirmed ESBL production [17]. Meropenem-resistant Gram-negative isolates were selected for further detection of MBL production with the disc potentiation method using imipenem (10 µg) and meropenem (10 µg), with and without EDTA (1 µg), as previously described [20].

### 2.4. DNA Extraction and Detection of bla_CTX-M_ and bla_VIM_ Genes

All phenotypically confirmed ESBL- and MBL-producing isolates were subjected to molecular detection of *bla*_CTX-M_ and *bla*_VIM_ genes using conventional PCR. The isolates were inoculated into 5 mL of Luria–Bertani broth (Hi-media, India) and incubated at 37 °C for 24 h. Following incubation, plasmid DNA was extracted using the alkaline lysis technique [21]. After extraction, DNA samples were suspended in 50 µL of TE buffer and kept at −20 °C. Genetic amplification was conducted in a 25 µL reaction volume containing 12.5 µL master mix (Solis Biodyne, Tartu, Estonia), 8.5 µL nuclease-free water, 3 µL of the plasmid DNA, and 0.5 µL each of forward (*bla*_CTX-M_: 5’-TTT GCG ATG TGC AGT ACC AGT AA-3’; *bla*_VIM_: 5’-GAT GGT GTT TGG TCG CAT A-3’) and reverse (*bla*_CTX-M_: 5’-CTC CGC TGC CGG TTT TATC-3’; *bla*_VIM_: 5’-CGA ATG CGC AGC ACC AG-3’) primers (Macrogen, Seoul, Korea) under the following optimal conditions: initial denaturation at 94 °C for 5 min, denaturation at 95 °C for 45 s of 35 cycles, annealing at 65 °C for 45 s of 35 cycles for *bla*_CTX-M_ and 56 °C for 45 s of 35 cycles for *bla*_VIM_, extension at 72 °C for 30 s of 35 cycles for *bla*_CTX-M_ and 72 °C for 45 s of 35 cycles for *bla*_VIM_, and final extension at 72 °C for 10 min. The amplified PCR products were separated on a 1.5% agarose gel in 1X TAE buffer (0.04 Tris-acetate, 0.001 M EDTA, pH 8.0), dyed with ethidium bromide, and observed with a gel-doc system. The amplicon sizes of *bla*_CTX-M_ and *bla*_VIM_ genes were 560 bp [22] and 390 bp [23], respectively.

### 2.5. Quality Control

Each batch of medium and reagents was subjected to sterility and performance testing. Duplicate culture was performed to ensure that growth was not due to contamination or any external source. During the antibiotic susceptibility testing, quality control was performed using control strains of *E. coli* ATCC 25922.

### 2.6. Data Analysis

All raw data obtained during the study period were tabulated using SPSS V.26. The chi-square test was used to draw associations between categorical variables, and a *p*-value ≤0.05 was considered to be a statistically significant association.

## 3. Results

### 3.1. Distribution of Bacterial Isolates

Bacterial colonization with at least one bacterium was seen in 25 (28.74%) stethoscopes. Among them, the majority were monobacterial contamination (17, 68%), whereas the other 8 (32%) showed polybacterial growth. The remaining 62 (71%) stethoscope samples did not exhibit any bacterial growth. In the 25 contaminated samples, a total of 34 bacterial isolates were recovered. Gram-positive bacteria accounted for 18 (52.94%) isolates, all of which were *Staphylococcus* spp. Of the remaining 16 (47.06%), Gram-negative bacilli from the genus *Acinetobacter* were the most frequent (7, 43.75%), followed by *Pseudomonas* spp. (5, 31.25%). *Citrobacter* spp. were encountered in only two (12.5%) stethoscopes, and *Escherichia coli* and *Klebsiella* spp. were isolated only once each (6.25%). Of the 18 staphylococci, the majority (83.33%) were *S. aureus*, and the remaining 16.77% were coagulase-negative staphylococci (CONS) (Figure 1).

### 3.2. Association of Different Variables with Colonization of Stethoscopes

A higher rate of contamination was observed in stethoscopes used by interns (6/15, 40.00%), whereas the least contamination was seen in stethoscopes used by doctors (5/31, 16.13%). However, contamination of stethoscopes was statistically not associated with the designation of the HCWs (*p* > 0.05). The majority of stethoscopes examined were those used by HCWs working in the outpatient department (OPD) (50/87, 57.47%). A higher rate of contamination was seen in stethoscopes used by HCWs from the inpatient department (IPD) (17/37, 45.95%) compared to those from the OPD (8/50, 16.00%), and this finding was statistically significant (*p* < 0.05). Only approximately half of the 52.87% HCWs (46/87, 52.87%) were found to be performing hand hygiene practices (HHPs) frequently. A higher rate of contamination was observed on stethoscopes used by HCWs who never performed HHPs (5/11, 45.45%), and the least contamination was found on stethoscopes used by HCWs who followed HHPs frequently (11/46, 23.91%). The stethoscope’s diaphragm (11/87, 12.64%) was considerably more contaminated compared to the earpieces (10/87, 11.49%) and bell (4/87, 4.59%), although the result was not statistically significant (*p* > 0.05). Different HCWs used different methods to cleanse their stethoscopes. Methylated spirit was used by most of the HCWs (61/87, 70.11%) to cleanse their stethoscope, and the rate of contamination was observed the least with its usage (14/61, 22.95%). The stethoscopes used by HCWs who never cleansed them with anything were more frequently contaminated (9/15, 60.00%). Statistically, we found that stethoscope contamination and disinfection techniques had a significant association (*p* < 0.05). The least contamination was seen on stethoscopes that were cleansed daily (4/27, 14.81%). Statistically, it was found that the rate of contamination was strongly associated with the frequency of cleansing (*p* < 0.05) (Table 1).

### 3.3. Antibiogram Pattern of the Isolates

Of the 34 isolates, 18 were Gram-positive and 16 were Gram-negative. Against the Gram-positive isolates, chloramphenicol (83.33%) was the most effective drug, followed by tetracycline and amikacin (72.22%). Azithromycin was the least effective drug (27.78%) for them, followed by ciprofloxacin (38.88%). The prevalence of methicillin-resistant *S. aureus* (MRSA) was eight (44.44%). Imipenem was the most effective drug against Gram-negative isolates (13, 81.25%) followed by ciprofloxacin (12, 75%). The least effective drug was cefepime (8, 50%). Approximately half of the Gram-negative isolates were MDR (16, 47.06%). The antibiotic susceptibility patterns of the isolates are presented in Table 2.

### 3.4. Prevalence of ESBL and MBL Producers and Presence of bla_CTX-M_/bla_VIM_ Genes

Phenotypic detection of ESBL production using the combined disc method revealed that eight isolates were screening positive, whereas only three isolates were MBL-producers on primary screening. Confirmatory testing for ESBL production showed seven positive isolates, of which five isolates showed the presence of *bla*_CTX-M_ gene. Similarly, confirmatory testing revealed that two isolates were MBL producers, one each of *Klebsiella* spp. and *Pseudomonas* spp.; however, the *bla*_VIM_ gene was detected only in *Pseudomonas* spp. (Table 3).

## 4. Discussion

This study revealed a low rate of stethoscope contamination (28.74%) compared to several similar studies previously performed in different hospitals [24,25,26,27,28]. HCW and staff medical devices have been shown to be potential carriers of pathogenic organisms in past studies [29]. Variations in the rate of stethoscope contamination in different hospitals can be ascribed to various factors, such as frequency of sample examination, exact use of the instrument, user’s dedication to hygiene, and frequency of disinfection, among others [30]. Only 34 bacterial isolates belonging to 7 different genera were recorded in our study. Gram-positive bacteria were slightly more prevalent (52.94%) than Gram-negative bacteria (47.06%). *Staphylococcus aureus* was the dominant isolate (83.33%) among Gram-positive bacteria. A similar study performed by Singh et al. in 2013 also reported *S. aureus* as the predominant isolate [28]. A study performed by Treakle et al. in 2008 revealed that *S. aureus* was the predominant bacterium in the white coats of medical staff at the Maryland Medical Center in Baltimore, Maryland [29]. These findings suggest that *S. aureus* is the most frequent bacteria in hospital settings and that it can be found in various fomites associated with doctors and nurses. In contrast to our findings, some studies have reported *Micrococcus* spp. as the dominant isolate on stethoscopes, but these bacteria were not observed in the current study [2,24]. Thapa and Sapkota reported a prevalence of only 3.94% Gram-negative bacilli in stethoscopes used by medical personnel at Chitwan Medical College, a hospital near our study hospital, which is very low compared to the current study [2]. These results indicate that contamination by Gram-negative bacteria is not as frequent as Gram-positive bacterial contamination of hospital equipment. The probable reason why Gram-positive bacteria, such as staphylococci and micrococci, account for a larger proportion is that they are the normal flora of the skin in humans and can easily be transferred to stethoscopes while examining patients [31]. In this study, the most contaminated part of the stethoscope was found to be the diaphragm, with a contamination rate of 12.64%, followed by the earpieces (11.49%) and bell (4.59%). A similar study in Chandigarh city in India also revealed that the diaphragm (53%) was the most contaminated part, as compared to the bell (21%) [26]. This pattern of the bacterial distribution of stethoscopes aligns with findings from an earlier study performed in Pokhara and Bharatpur in Nepal [2,32]. Such a pattern of bacterial growth on stethoscope parts is to be expected because the diaphragm comes in direct contact with the patient’s skin, while the earpieces and bell are associated with the commensals of HCWs. In addition, the surface area of the diaphragm is larger compared to the other parts of the stethoscope, which increases the likelihood of bacterial contamination [33].

The current investigation determined a higher rate of bacterial contamination on stethoscopes used by interns (40%), followed by nurses (34.15%) and doctors (16.13%). This finding contradicts the results of similar studies by Bhatta and Datta separately, which showed that doctors and nurses had more contaminated stethoscopes than any other groups [26,32]. There can be considerable variation in this finding from hospital to hospital, as stethoscope contamination largely hinges on the individual’s personal hygiene and method of cleansing the equipment. However, it cannot be denied that, since interns are still in the learning phase, they may forget to cleanse their personal equipment or sometimes deliberately ignore personal hygiene. [34]. The present study revealed that the rate of stethoscope contamination was higher among HCWs working in the IPD (45.95%) compared to stethoscopes used by HCWs working in the OPD (16%), and this finding was statistically significant. An earlier study performed in Pune, India, reported a similar result, indicating a higher rate of stethoscope contamination among IPD staff [35]. However, just the opposite finding was made in a few other studies performed in India, where researchers discovered a higher rate of contamination among OPD staff [27,28]. IPD staff mostly encounter and examine hospitalized patients and thus there might be a chance that their personal equipment, including stethoscopes, will be contaminated more often [24]. Contaminated stethoscopes pose a threat to the health of patients, especially immunocompromised patients, as they frequently visit hospitals [36]. Furthermore, the incidence of *Staphylococcus aureus* in the ICU and *Escherichia coli* and *Klebsiella* in surgical sites raises a serious question about the hospital’s SOP and sanitation level. Lower bacterial contamination was found on the stethoscopes of HCWs who practiced hand hygiene after examining every patient compared to those who never followed hand hygiene practices after touching patients (23.91% vs. 44.45%, respectively), while those who seldom practiced hand hygiene had a 30% contamination rate. This finding matches the findings from a study performed by Dagnaw in Ethiopia [25]. The World Health Organization (WHO) has stated that hand hygiene is fundamental in ensuring patient safety and should be performed timely and effectively in the process of care [37]. Health professionals were found to use different methods of cleansing/disinfecting their stethoscopes. The most effective disinfectant was hand sanitizer, with 87.5% efficacy, followed by methylated spirit swabs, with 77.04% efficiency, while cleansing the stethoscope with cloths was only 66.67% effective. We noticed that approximately half of the stethoscopes that were never disinfected or cleansed were contaminated. Our results are consistent with findings from a study performed by Bhatta et al. in 2011, in which the disinfection rate of spirit swabs was found to be 74.13% [24]. In contrast, our findings contradict a study performed in Chitwan, Nepal, which showed that the effectiveness of methylated spirit swabs was 57.2%, and hand sanitizer had only 23.9% efficacy [2]. The different efficacies of alcohol-based hand sanitizers (70% alcohol) and methylated spirit swabs (95% alcohol) can be explained by the fact that concentrated alcohol evaporates faster, thus reducing the time for penetration and destruction of the cell wall of microorganisms [37]. The frequency of stethoscope disinfection/cleansing was largely found to affect bacterial contamination. In our study, 60.00% of the stethoscopes that were never cleansed were found to be contaminated, whereas 38.46% of the stethoscopes disinfected only once a month were found to be contaminated, and the rate of contamination decreased gradually as the frequency of cleansing increased, as only 14.81% of stethoscopes were contaminated when they were cleansed daily. A similar study performed in Nigeria by Uneke et al. reported that the rate of contamination of stethoscopes decreased as the frequency of cleansing increased [38]. If stethoscopes are used without being disinfected following every patient examination, there is no denying that the bacterial flora from patients will contaminate the apparatus, which may increase the burden of nosocomial infections.

Looking at the pattern of antimicrobial susceptibility, we can see that bacteria are increasing their resistance against commonly prescribed drugs day by day. Chloramphenicol (83.33%) was the most effective drug against Gram-positive bacteria. Similar research performed by Dagnaw in Ethiopia showed that vancomycin was the most effective drug for Gram-positive isolates [25]. A previous study conducted at Chitwan Medical College in Bharatpur, Nepal, found vancomycin to be the most effective drug, with 95.7% efficacy [2]. In contrast, a study conducted in Nigeria showed that chloramphenicol was resisted by all Gram-positive isolates. In the case of Gram-negative bacteria, imipenem was the most effective drug (81.25%) in the present study. Imipenem was also found to be the most effective drug for Gram-negative bacteria in an earlier study performed by Bhatta et al. in 2018 [32]. We discovered that approximately half (47.06%) of the Gram-negative isolates were MDR, and the prevalence of MRSA was approximately 45%. In the current study, phenotypic screening demonstrated that 43.75% of the Gram-negative isolates were ESBL producers, and 18.75% were MBL producers. In a similar study conducted in Bangladesh to investigate bacterial contamination of stethoscopes, only 1% of isolates were found to be ESBL-producers, whereas 12% were MRSA [39]. Meanwhile, a similar study performed in India reported 6.4% ESBL-producing Gram-negative bacteria on stethoscopes, while the prevalence of MRSA was 25% [26]. Previously, in 2018, a group of researchers in Iran found that 20.6% of isolates recovered from hospital equipment were MBL-producers, which is comparable to our study. In the current study, *bla*_CTX-M_ and *bla*_VIM_ genes were detected in 31.25% and 6.25% of isolates, respectively. In his study, Kasim reported that approximately 55.2% of isolates from stethoscopes were carriers of the *bla*_CTX-M_ gene [40]. Meanwhile, none of the seven MBL-producing isolates recovered from hospital devices in a 2018 study in Iran by Moghadampour et al. possessed the *bla*_VIM_ gene [41], whereas only one MBL-producing isolate was found to be carrying the *bla*_VIM_ gene in the current study. The prevalence of antibiotic-resistant genes such as *bla*_VIM_ in bacteria in important equipment like stethoscopes is a red flag, as it can easily be transmitted from and within the hospital setting [42]. Although the prevalence of ESBL- and MBL-producing organisms on stethoscopes is variable in different studies, it is alarming that even a small number of these organisms residing on such crucial equipment makes stethoscopes a perfect vector for bacterial transmission [43]. Another problem associated with such bacteria is that the plasmid-mediated resistance mechanism is easily transferrable to other bacteria, making the latter drug-resistant [26].

## 5. Conclusions

More than a quarter of the stethoscopes examined were colonized by pathogenic bacteria. Bacterial contamination of stethoscopes was found to be significantly associated with attributes such as disinfecting routine, method of disinfection, and department of the hospital. Detection of MDR, as well as ESBL- and MBL-producing bacteria, in daily and frequently used medical devices such as stethoscopes calls for immediate interventions by hospitals. Further research should encompass other contaminating organisms, such as anaerobic bacteria, fungi, and viruses, to explore their role as nosocomial pathogens.

## Figures and Tables

**Figure 1 diseases-11-00055-f001:**
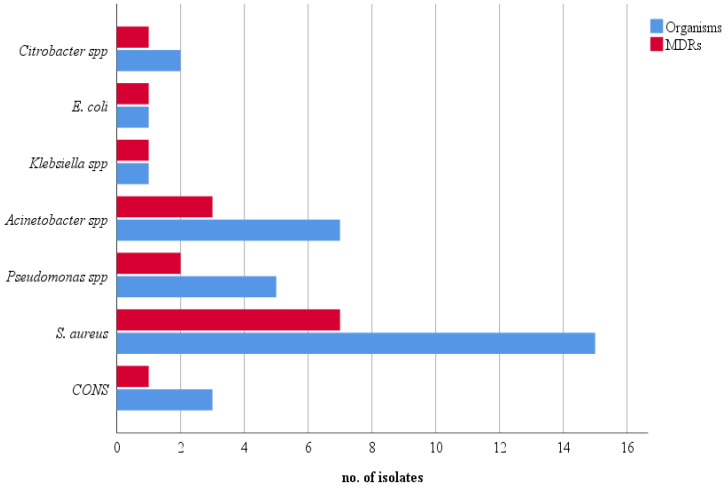
Frequency of total and MDR isolates.

**Table 1 diseases-11-00055-t001:** Association of different variables with contamination of stethoscopes.

Attributes	Parameters	Samples	Growth	*p*-Value
Designation of staff	Doctor	31	5 (16.13%)	0.351
	Nurse	41	14 (34.15%)	
	Intern	15	6 (40%)	
Department	OPD	50	8 (16%)	0.006 *
	IPD	37	17 (45.95%)	
Hand hygiene practice	Frequently	46	11 (23.91%)	0.359
	Never	11	5 (45.45%)	
	Sometimes	30	9 (30%)	
Stethoscope parts	Diaphragm	87	11 (12.64%)	0.149
	Ear piece	87	10 (11.49%)	
	Bell	87	4 (4.59%)	
Disinfectant used	Cloth	3	1 (33.33%)	0.026 *
	Hand sanitizer	8	1 (12.5%)	
	Methylated sprit	61	14 (22.95%)	
	Never disinfected	15	9 (60.00%)	
Frequency of cleansing	Everyday	27	4 (14.81%)	0.021 *
	Alternate days	18	3 (16.67%)	
	Once a week	14	4 (28.57%)	
	Once a month	13	5 (38.46%)	
	Never	15	9 (60.00%)	

* Significant at 5% level of significance.

**Table 2 diseases-11-00055-t002:** Antibiogram of the isolates.

	Gram-Positive Isolates		Gram-Negative Isolates	
Antibiotics	S (%)	R (%)	S (%)	R (%)
Tetracycline	13 (72.22%)	5 (27.78%)	-	-
Imipenem	-	-	13 (81.25%)	3 (18.75%)
Chloramphenicol	15 (83.33%)	3 (16.64%)	1 (50.00%)	1 (50.00%)
Ciprofloxacin	7 (38.88%)	11 (61.12%)	12 (75%)	4 (25%)
Gentamycin	12 (66.67%)	6 (33.33%)	12 (75.00%)	4 (25.00%)
Azithromycin	5 (27.78%)	13 (72.22%)	-	-
Methicillin	10 (55.56%)	8 (44.44%)	-	-
Ceftazidime	-	-	11 (68.75%)	5 (31.25%)
Cefotaxime	6 (33.33%)	12 (66.67%)	8 (50.00%)	8 (50.00%)
Cefepime	5 (62.5%)	3 (37.5%)	8 (50.00%)	8 (50.00%)
Amikacin	13 (72.22%)	5 (27.78)	-	-
Aztreonam	-	-	9 (56.25%)	7 (43.75%)
Levofloxacin	-	-	10 (90.91%)	1 (9.09%)

- = Not Tested.

**Table 3 diseases-11-00055-t003:** Distribution of ESBL- and MBL-producing isolates.

		Screening Test Positive		Confirmatory Test Positive		Detection of Gene	
Organism	Significant Growth	ESBL	MBL	ESBL	MBL	*bla*_CTX-M_ Gene	*bla*_VIM_ Gene
*Pseudomonas* spp.	5	2	1	2	1	1	1
*Acinetobacter* spp.	7	3	0	3	0	2	0
*Escherichia coli*	1	1	1	1	0	1	0
*Klebsiella* spp.	1	1	1	1	1	1	0
*Citrobacter* spp.	2	1	0	0	0	0	0
Total	16	8	3	7	2	5	1

## Data Availability

The data used to support the findings of this study are included within the article.

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
