# Peer review of "Bacterial Assessment of Stethoscopes Used by Healthcare Workers at a Tertiary Care Government Hospital in Bharatpur, Nepal"

_diseases, 2023, doi:10.3390/diseases11020055_

Round 1

Reviewer 1 Report

The article is well-structured, the design of the studio is appropriate and the topic covered is of considerable importance. Although the subject has already been analysed in other studies, the local situation in Nepal is adequately analysed in this study. The results were well presented and the discussions well argued. The conclusions, while confirming what is known in the literature, bring a useful feedback of the Nepalese reality. In my opinion the article does not need revisions.

Reviewer 2 Report

Bacterial Assessment of Stethoscope Used by Health-Care Workers at a Tertiary-Care Government Hospital in Bharatpur, Nepal

Thank you for presenting this manuscript focusing on the issue of the prevalence of contaminated of stethoscopes in hospitals. It is an important topic, which is usually not focused. The manuscript is well-drafted and to the mark however, there are a few points that must be addressed. Please see below.

1.     In abstract, a sentence states- “… amongst the HCWs who overlooked the hand hygiene practices (45.45%).” What was the percentage of such HCWs? That result is not presented the in abstract.

2.     Page 5- what is the denominator for the samples taken? What is the total number of possible participants?

3.     The contamination rate is far lower than in other studies where it was recorded 86% to 89%.      Jain A, Shah H, Jain A, Sharma M. Disinfection of stethoscopes: Gap between knowledge and practice in an Indian tertiary care hospital. Ann Trop Med Public Health 2013;6:236-9

Where do you keep your study results and why do you think you got this lower percentage?

4.     How generalizable are the results?

5. The reasons stated for low rates are- “… frequency of sample examined, the exact use of the instrument (eg, taking blood pressure, auscultation of the lungs, heart, abdomen, or great vessels), or the frequency of disinfecation”.

       Do you see ‘the frequency of the sample examined’ in your study, as a limitation?

6.      Please rewrite this sentence. – “However, as the interns are in the learning phase, therre may be a chance that they may be oblivious of the importance of cleansing their personal equipments”

7.    Font sizes are different in different parts of the manuscript.

8.   There are some typing or spelling errors such as ‘there’ is written as ‘therre’ on Page 7 (sentence written above).

9  The next sentence says- “The present revealed that the rate …” a connecting word is missing in this sentence.

10.   Correct the title of reference 27.

11.   Page 8- “We noticed that around half of the stethoscpes who were never disinfected nor cleansed were contamined.” It is noteworthy that half were not contaminated despite of not being cleaned at all. Comment on this too.

12.   The following statement is written several times- “… the rate of contamination of stethoscope decreases as the frequency of cleansing increases is repeated several times”.  which is a well-established relation. Please make this brief. What do you want to implicate? 

Reviewer 3 Report

The article by Sanjib Adhikari et al. describes the bacteriology of the widely used hospital equipment, the stethoscope, in a Hospital in Nepal.

Global: The manuscript is overall well written but I would recommend a rework of the numberuss typos/font changes. Also, bacteria and "et al." should be written in italics and it is better to use passive forms.

Method: How was the number of subjects/stethoscope to be included determined?

Method: Describe more precisely the transport media, the plating methods and media, the reading times, the operators involved in each step...

Results: A table summarizing stethoscope selection and owner demographics/characteristics seems important

Results: A distribution of the number of strains per stethoscope seems interesting.

Global: Acronyms should be introduced before their first use.

"Significant at 5% level of significance" should be indicated in the methods and not in the results (table 1)

Reviewer 4 Report

ID: diseases-2200632

Title: Bacterial Assessment of Stethoscope Used by Health-Care Workers at a Tertiary-Care Government Hospital in Bharatpur, Nepal

Thank you for providing a chance to review this manuscript.

Comment: Reject.

Detailed information:

Abstract

1) Why “Medical devices like” and “In addition, molecular detection of” are different from other words? Please standardize the font and size throughout the article. The manuscript is reflecting the authors' attitude. That is to say, I cannot see how serious and meticulous the authors were in this blind-review manuscript.

2) How did you assess participants who have “overlooked the hand hygiene practices”? A questionnaire or a scale? Please explain briefly.

3) What is the full name of “ESBL”? The full name is required for the first appearance of the abbreviation in either the abstract or the text.

4) I don't understand the meaning of “the judicious use of antibiotics”. Is it for stethoscope use or for patient use? Is that an appropriate conclusion? As far as I know, multi-drug resistant (MDR) strains often arise because of the misuse of antibiotics.

1. Introduction

Paragraph 2, page 2: It is not necessary to list such a detailed list of bacterial pathogen types, but a categorical description is sufficient. Additionally, I think the authors should have elaborated more on the potentially serious consequences of stethoscope contamination rather than just one sentence.

2. Methodology

2.2. Antibiotic susceptibility testing, page 3: There seems to be a problem with the formatting here, please modify it.

3. Results

3.2. Association of different variables with colonization of stethoscopes, page 5: What are “OPD” and “IPD”?

Table 3, page 6: I think the black line above the "Total" line can be removed.

4. Discussion

1) The format and font of the entire “Discussion” section is very messy and inconsistent, so please double-check and modify it. The content of each paragraph is long and mixed up, and it is impossible to see the central idea that is intended to be expressed in it. Please sort out the logic of the article carefully and then express it in a reasonable and clear way.

2) I notice that this section lists the results extensively, which I think is unnecessary. Information that has already appeared in the “Results” section does not need to be repeated, unless the author considers it to be very important, it should only be stated in words.

3) Please add the strengths, limitations, and future research directions of this study.

I did not see the need or innovation in this article. First, the argument that stethoscope contamination may lead to patient exacerbation is not reflected in the study. Secondly, the results of the study showed a lower contamination rate than previous studies, whether this was related to the sterilization measures or management practices adopted by the hospital was not stated in the article. Finally, I believe the authors should have suggested more specific improvements rather than just calling on hospitals to take measures.

Thank you and my best,

Your reviewer

Round 2

Reviewer 3 Report

Manuscript has been well improved.

Reviewer 4 Report

ID: diseases-2200632

Title: Bacterial Assessment of Stethoscope Used by Health-Care Workers at a Tertiary-Care Government Hospital in Bharatpur, Nepal

Thank you for providing a chance to review this manuscript.

Comment: Minor revision.

Detailed information:

1. Introduction

Paragraph 2, page 2: About the content of this study, I think it is possible to start a separate paragraph and add more descriptions, such as the study hypothesis. Remember, there is only one central idea in a paragraph, do not let one paragraph express more than one topic.

3. Results

3.2. Association of different variables with colonization of stethoscopes, page 5: What are “OPD” and “IPD”? I saw the authors' response, and although it is common terminology, I still maintain the suggestion of adding its full name when it first appears as an acronym. This will be convenient for readers.

Table 3, page 6: I think the black line above the "Total" line can be removed.

4. Discussion

The authors still did not add the limitations of this study, which I think is a very necessary point.

The overall quality of the article has improved, but there are still some parts that need to be modified. In particular, in my opinion, too long for every paragraph of Discussion section, and appropriate changes to the paragraph and sentence structure would make the essay more readable.

Thank you and my best,

Your reviewer
